# Sperm whales habituate to research vessels engaged in photoidentification

**Hal Whitehead**◉*, **Christine M. K. Clarke**◉, **Ana Eguiguren**

Department of Biology, Dalhousie University, Halifax, Nova Scotia, Canada

* hwhitehe@dal.ca

## Abstract

Wildlife research has the potential to affect the subject animals. These effects can be problematic from ethical and management perspectives, as well as for the effectiveness of the research process itself. For individually-based behavioral research, repeated identifications are often necessary, which can lead to habituation or sensitization as animal responses to the researchers change over time because of learning. Sperm whales (*Physeter macrocephalus*) photoidentified off the Galápagos Islands became more tolerant of vessels engaged in photoidentification as they gained more experience of the vessels (as indicated by reductions in both the proportion of poor photographs taken and the range to the photographed whales). Thus, the whales habituated to the photoidentification process, making it more effective and suggesting little deleterious impact. Studies of habituation or sensitization should be carried out when wild animals are repeatedly exposed to disturbance from research activities.

## Introduction

Animals often react behaviorally to human intrusions into their space and habitat. When the intrusion is directed towards the animals, as in hunting, ecotourism or research, these behavioral changes may be significant both for the animals and the objectives of the human activity. As the goal of much wildlife research is to protect species and their habitat, deleterious effects of the research process are particularly problematic from ethical and management perspectives [1]. Changes in behavior in response to the research may also affect the efficiency of the research process itself, either positively or negatively [2,3].

The effects of the human intrusion can change with exposure. With repeated or prolonged intrusions, animals can become habituated ("the relative persistent waning of a response as a result of repeated stimulation…" [4]), or they can be sensitized ("increased behavioral responsiveness over time…" [5]), to the stimulus. Bejder et al. [6] emphasize that that a habituation-type response (*i.e.*, declining with time) is not necessarily true habituation, nor an indicator that there has been no negative impact

**Data availability statement:** All relevant data are within the manuscript and its Supporting Information files.

**Funding:** The funders (Natural Sciences and Engineering Research Council of Canada, Whale and Dolphin Conservation Society, National Geographic Society, International Whaling Commission, Cetacean Society International, Green Island Foundation) had no role in study design, data collection and analysis, decision to publish, or preparation of the manuscript.

**Competing interests:** The authors have declared that no competing interests exist.

of the disturbance. For instance, animals could be deafened by an acoustic disturbance and so reduce their behavioral response to it.

Research activities on marine mammals span the gamut from lethal sampling to passive acoustic monitoring from moored or drifting hydrophones, which likely have no meaningful effect on the animals. There have been a number of studies of the impacts of some common field techniques such as biopsy sampling [7], satellite tagging [e.g., 8], and drone photography [e.g., 9]. Photoidentification of individuals, one of the most commonly used and informative field techniques for wild cetaceans [10], has received hardly any research on potential impacts, despite calls for such studies [1]. Although individual identification photographs of cetaceans can sometimes be taken from land [11] or aerial vehicles [e.g., 12], the vast majority are obtained from vessel-born cameras, and these ships or boats may disturb the animals, especially when they maneuver to obtain a useful range or orientation for photoidentification.

Here, we examine progressive changes in the behavior of sperm whales (*Physeter macrocephalus*) to research vessels primarily engaged in photoidentification off the Galápagos Islands. This data set has several features that make it suitable and interesting. Field seasons lasted several months within which individually-identified sperm whales were repeatedly exposed to the research vessel. Vessel traffic is light in the deeper waters off the Galápagos that the sperm whales use, and the whales are very unlikely to have been exposed to similar vessels engaged in similar behavior in the area, prior to our research. The earliest study of our series was 1985, just four years after the end of intense commercial whaling off neighbouring Perú [13], an area well within the range of Galápagos animals [14], so some of the whales that we followed may have had fairly recent intense experience with different types of vessels operated with very different goals. In the North Pacific during the early 19th Century, sperm whales responded swiftly and effectively to a novel and dangerous anthropogenic change to their environment, the open-boat whaler [15]. This study examines potential changes in sperm whale behavior to a human intrusion which is more benign than whaling but not necessarily harm free.

## Methods

### Data

The primary goal of the field research projects was to collect data on the social structure of female sperm whales. This required repeated identifications of individual animals in order to measure dyadic associations and build models of social structure [see 16]. Data were collected from 10-m, 12-m and 14-m auxiliary sailing vessels in the waters (usually >1,000m deep) off the Galápagos Islands during the primary study years 1985, 1987, 1989, 1995 and 1999, as well as during shorter studies between 1988–1998 [Table 1; supplementary material for 17]. Although the Galápagos studies continued into the 21st century, the technological development of photoidentification techniques, especially the use of digital cameras with increasing resolution, invalidated employing the methods used in this analysis on those data.

Groups of whales (mostly containing between about 8–40 animals) were tracked using passive acoustics and visually (in daytime) for periods of 1–12 consecutive

**Table 1. Primary studies of sperm whales off the Galápagos Islands used in the analysis.**

| Year | Days with whales | Days with scan samples | First day | Last day |
| --- | --- | --- | --- | --- |
| 1985 | 31 | 31 | 23-Feb | 20-Apr |
| 1987 | 54 | 54 | 03-Jan | 28-Jun |
| 1989 | 56 | 0 | 13-Jan | 21-May |
| 1995 | 16 | 0 | 27-Apr | 03-Jun |
| 1999 | 18 | 0 | 10-Mar | 12-Apr |

days [18]. The groups consisted of one or more matrilineally-based, nearly permanent, social units of about ten female sperm whales and their young [19,20]. Group membership typically changed every day or two as units joined and left the group [18]. During daytime, photographs were taken of the flukes (tail) of the whales usually at the start of deep dives (fluking up). The marks on the flukes shown in the photographs were used to identify individuals [21]. Each photograph was given a quality value ranging from $Q=1$ (extremely poor) to $Q=5$ (excellent) based on focus, size, orientation proportion visible, and image exposure [Fig 1; 21]. Photographs were matched within and between years to produce a catalog of Galápagos sperm whales [18]. Photographs of mature males, which accompanied the groups of females for periods of hours, and calves, distinguishable by their large and small sizes respectively and with distinctively different behavior, were excluded from this analysis.

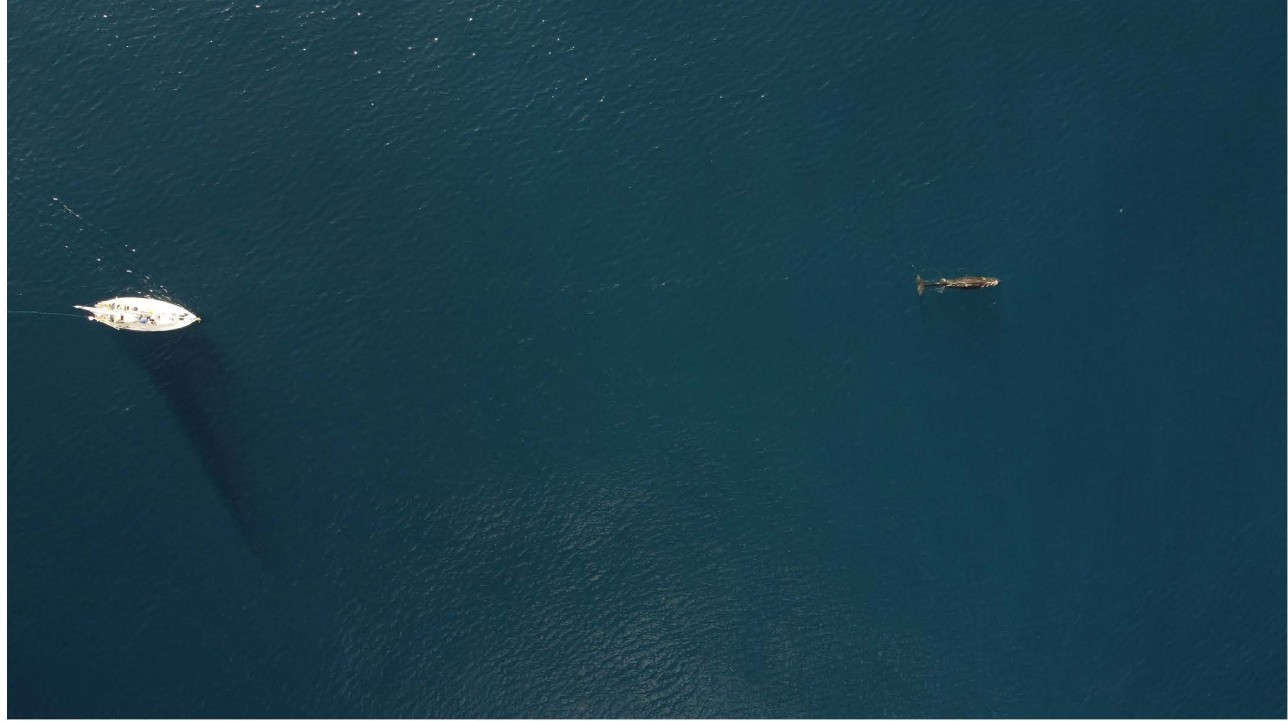

**Fig 1. Following a sperm whale (~10m long) from the research vessel (12m long) to obtain a fluke photograph.** This image is the property of the authors of this manuscript.

During these studies, the general protocol was for the research vessel to approach whales breathing at the surface between their deep dives under engine, staying about 50m directly behind them, matching their speed (about 2 knots) and changing direction and speed as little as possible (see Fig 2). Thus, we tried to take identification photographs with little disturbance to the whales. There was never any attempt to biopsy or tag the whales. However, sometimes whales would seem to try to avoid the research vessel (by behavior such as changing their direction frequently, speeding up, or making shallow dives so their flukes were not well shown), leading to photographs at greater ranges, poor angles, and so with lower $Q$ values.

This analysis used study years with more than 15 days spent following whales and during which photoidentifications were collected with Canon SLR 35mm film cameras and 300mm lenses: primary study years (Table 1).

Additionally, during 1985 and 1987 there were consistent scan samples of visible whales collected every 5 minutes during daylight. These samples included estimates of range to the whales fluking up (in metres), and records of whether a photograph was taken of the fluke [22].

Ethics statement: Field work was carried out under the auspices of the Charles Darwin Research Station, with the permission of the Galápagos National Parks Service, the Ecuadorean Instituto Nacional de Pesca, and the Armada of Ecuador.

## Data analysis

The unit of analysis was a day spent following whales (as in columns 2 and 3 of Table 1). The photoidentification and scan-sample data were used to calculate two principal dependent variables, three independent variables, and two control variables for each day.

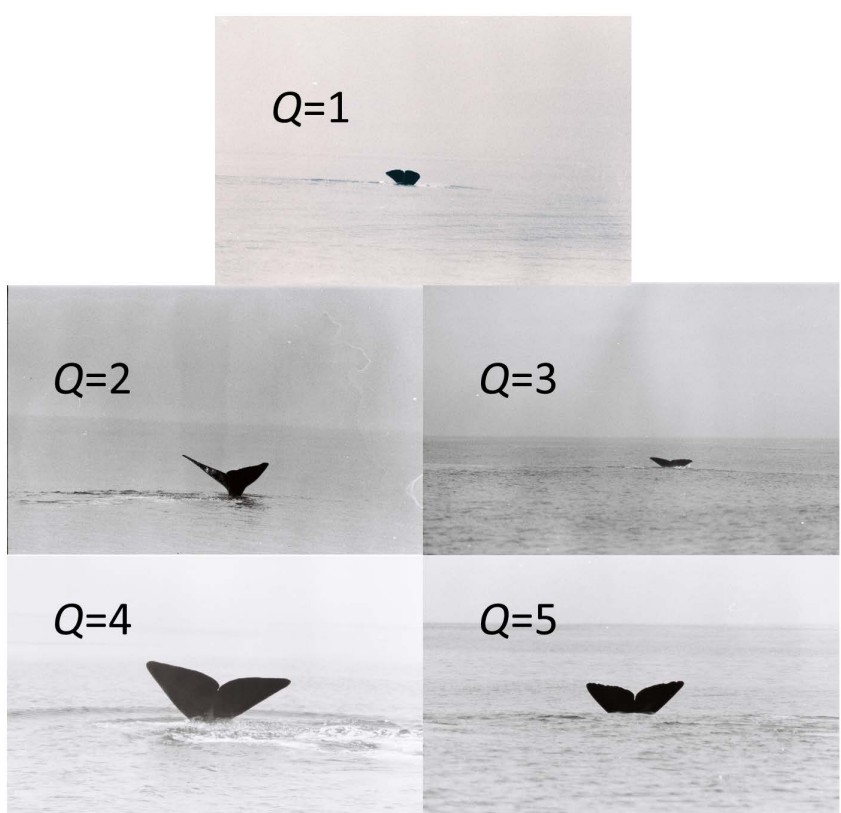

**Fig 2.  $Q$ values of photoidentifications from 24 Feb 1985, 10:45-16:10 local time, except Q=3 (7 March 1985, 08:45).** Images are the property of the authors of this manuscript.

### Dependent variables

The two principal dependent variables were:

$q_{13}$ Proportion of photoidentifications on a day that were of poor quality: $Q = 1–3$.

$r_{med}$ Median estimated range (in m) to photographed flukes (collected during scan samples just in 1985 and 1987).

These dependent variables indicate the whales' discomfort (lack of tolerance) with the research vessel, and thus the efficiency of the research. If the animals try to evade the research vessel by changing direction, shallow diving, or swimming faster, photographs will generally be of poorer quality (*e.g.,* smaller images of flukes, poorer angle or focus) or further away.

The analyses were also carried out on two additional dependent variables: $q_{12}$ the proportion of photoidentifications on a day that were very poor: $Q = 1–2$; and $r_{mean}$ the mean estimated range (in m) to photographed flukes (just 1985 and 1987). However, the results from $q_{12}$ were generally similar to those for $q_{13}$ and less powerful (as the numerators in the proportions were smaller, and thus $q_{12}$ was less precise than $q_{13}$), and those from $r_{mean}$ were similar to those from $r_{med}$. The results from these additional dependent variables are summarized in S1 Supporting Information.

### Independent variables

*api*. Whether any animals photoidentified on that day that had been photoidentified previously (binary measure). This indicates whether there was any familiarity with the research vessels by the whales being followed.

*ppi*. Proportion of animals photoidentified on that day that had been photoidentified previously. This indicates the level of familiarity with the research vessels among the whales being followed.

*cpi.* Cumulative number of previous days that animals photoidentified on that day had been photoidentified (so if the 10 animals were photographed on the focal day had been photographed on 3, 0, 6, 8, 0, 0, 1, 4, 2, and 1 previous days, *cpi* = 25). This measure summarizes the amount of previous exposure to the research vessels by the whales being followed.

### Control variables

*Year* (categorical random variable). Photoidentification and *Q*-rating methods, as well as whale behavior, potentially changed between study years.

*sdd* Number of days since the start of field season, thus controlling for systematic changes in photoidentification methods or crew abilities or whale behavior as field seasons progressed.

Analyses were repeated with *api, ppi* and *cpi* calculated just considering identifications of each individual in the primary study year, ignoring identifications in previous years. These analyses model the possibility of little or no between-year memory of encountering the research vessel.

The analyses examined the effects of each of the independent variables (*api, ppi, cpi*) which quantify the previous experience to the research vessel of the group being followed, on each of the dependent variables ($q_{13}$, $r_{med}$), which measure the reactions of the group being followed to the presence of the research vessel, controlling for different study years (*Year*) and the number of days since the start of that year's field season (*sdd*). Correlations between *sdd* and each of the independent variables were small ($|r| < 0.23$), so there was no collinearity. The models fitted included simplified versions without independent and/or control variables, as well as quadratic versions of the independent variables *ppi* and *cpi*, and the control variable *sdd*. We fitted generalized linear mixed effects models with *Year* as a categorical random variable using the binomial distribution for $q_{13}$ as this is a proportion, and the normal distribution for $r_{med}$. Model fits were compared using AIC (Aikaike Information Criterion). The best-fitting model is indicated by minimum AIC, with discrepancies (ΔAIC) of less than 2.0 from the best-fitting model indicating models with some support [23].

# Results

The control variable $sdd$, the number of days since the start of field season, was included as a quadratic term in the best fitting models for both dependent variables. The importance of $sdd$ was relatively small compared with each of the independent variables for $q_{13}$, the proportion of poor-quality photoidentifications (Table 2). However, on its own, $sdd$ explained $r_{med}$, the median range, almost as well as the independent variables that measure whale experience of the research vessel

**Table 2. Fit of generalized linear and quadratic models to dependent variables (Proportion of poor photographs, $q_{13}$, and Median range to photographed animals, $r_{med}$) using different combinations of independent and control variables.**

| Dependent variable: | Proportion poor photographs (Q=1–3) | | | | Median range to photographed animals | | | |
|---|---|---|---|---|---|---|---|---|
| Model | AIC | ΔAIC | Coeff | *P* | AIC | ΔAIC | Coeff | *P* |
| 1 | 1515.30 | 1029.83 | | | 824.63 | 3.62 | | |
| $sdd$ | 1416.62 | 931.14 | | | 821.40 | 0.40 | | |
| $sdd + sdd^2$ | 1376.66 | 891.19 | | | 821.92 | 0.91 | | |
| $1 + (1\|Year)$ | 550.09 | 64.62 | | | 828.63 | 7.62 | | |
| $sdd + (1\|Year)$ | 546.40 | 60.93 | | | 825.40 | 4.40 | | |
| $sdd + sdd^2 + (1\|Year)$ | 533.37 | 47.90 | | | 825.92 | 4.91 | | |
| $api$ | 1516.01 | 1030.54 | −0.118 | 0.253 | 824.29 | 3.28 | −13.996 | 0.132 |
| $api + (1\|Year)$ | 543.47 | 57.99 | −0.222 | 0.046 | 828.29 | 7.28 | −13.996 | 0.127 |
| $api + sdd$ | 1418.60 | 933.13 | −0.007 | 0.000 | 821.01 | 0.00 | −0.137 | 0.024 |
| $api + sdd + (1\|Year)$ | 541.96 | 56.49 | −0.001 | 0.092 | 825.01 | 4.00 | −0.137 | 0.022 |
| $api + sdd + sdd^2$ | 1378.51 | 893.03 | −0.021 | 0.000 | 821.39 | 0.38 | 0.153 | 0.526 |
| $api + sdd + sdd^2 + (1\|Year)$ | 529.72 | 44.24 | −0.010 | 0.000 | 825.39 | 4.38 | 0.153 | 0.516 |
| $ppi$ | 1511.36 | 1025.89 | 0.204 | 0.015 | 825.92 | 4.92 | −8.616 | 0.408 |
| $ppi + ppi^2$ | 1480.93 | 995.46 | | | 827.92 | 6.91 | | |
| $ppi + (1\|Year)$ | 498.62 | 13.15 | −0.715 | 0.000 | 829.92 | 8.92 | −8.616 | 0.402 |
| $ppi + ppi^2 + (1\|Year)$ | 501.04 | 15.57 | | | 831.92 | 10.91 | | |
| $ppi + sdd$ | 1418.26 | 932.79 | −0.007 | 0.000 | 821.79 | 0.78 | −0.151 | 0.015 |
| $ppi + ppi^2 + sdd$ | 1413.98 | 928.51 | | | 823.61 | 2.61 | | |
| $ppi + sdd + (1\|Year)$ | 492.85 | 7.38 | −0.002 | 0.011 | 825.79 | 4.78 | −0.151 | 0.014 |
| $ppi + ppi^2 + sdd + (1\|Year)$ | 495.22 | 9.75 | | | 827.61 | 6.61 | | |
| $ppi + sdd + sdd^2$ | 1378.65 | 893.18 | −0.021 | 0.000 | 822.42 | 1.42 | 0.118 | 0.627 |
| $ppi + ppi^2 + sdd + sdd^2$ | 1376.38 | 890.91 | | | 824.36 | 3.36 | | |
| $ppi + sdd + sdd^2 + (1\|Year)$ | 485.47 | 0.00 | −0.010 | 0.001 | 826.42 | 5.42 | 0.118 | 0.619 |
| $ppi + ppi^2 + sdd + sdd^2 + (1\|Year)$ | 488.01 | 2.54 | | | 828.36 | 7.36 | | |
| $cpi$ | 1505.47 | 1020.00 | 0.003 | 0.001 | 826.15 | 5.15 | −0.129 | 0.497 |
| $cpi + cpi^2$ | 1506.98 | 1021.50 | | | 828.13 | 7.13 | | |
| $cpi + (1\|Year)$ | 491.70 | 6.22 | −0.009 | 0.000 | 830.15 | 9.15 | −0.129 | 0.492 |
| $cpi + cpi^2 + (1\|Year)$ | 494.31 | 8.83 | | | 832.13 | 11.13 | | |
| $cpi + sdd$ | 1417.04 | 931.57 | −0.006 | 0.000 | 821.77 | 0.76 | −0.156 | 0.013 |
| $cpi + cpi^2 + sdd$ | 1418.92 | 933.45 | | | 823.76 | 2.75 | | |
| $cpi + sdd + (1\|Year)$ | 487.59 | 2.12 | −0.002 | 0.028 | 825.77 | 4.76 | −0.156 | 0.012 |
| $cpi + cpi^2 + sdd + (1\|Year)$ | 489.83 | 4.36 | | | 827.76 | 6.75 | | |
| $cpi + sdd + sdd^2$ | 1375.48 | 890.01 | −0.021 | 0.000 | 822.25 | 1.25 | 0.126 | 0.603 |
| $cpi + cpi^2 + sdd + sdd^2$ | 1377.21 | 891.74 | | | 824.15 | 3.14 | | |
| $cpi + sdd + sdd^2 + (1\|Year)$ | 486.86 | 1.39 | −0.006 | 0.051 | 826.25 | 5.25 | 0.126 | 0.595 |
| $cpi + cpi^2 + sdd + sdd^2 + (1\|Year)$ | 489.06 | 3.58 | | | 828.15 | 7.14 | | |

(Table 2). Given the experience of the whales to the research vessel, the effectiveness of the photoidentification process (in terms of the proportion of good quality photoidentifications) seems to generally increase during the first 2–3 months of a field season, and decline thereafter (S1 Supporting Information Fig A1).

The table gives for each dependent variable, and combination of independent and control variables AIC, ΔAIC, the coefficient of the independent variable when it was fitted as a linear term, and its P-value. For each dependent variable, the best fitting model (minimum AIC) is highlighted in green, and other models with substantial support (ΔAIC <2.0) are highlighted in yellow.

For models of $q_{13}$, the proportion of poor photographs taken on a day, those with *Year* as a random control variable fit substantially better than the equivalent model without a *Year* effect (ΔAIC>>10 in all cases; Table 2). $q_{13}$ was best explained by a decreasing linear function of *ppi* (the proportion of animals that had been photoidentified previously), although a decreasing function of *cpi* (the cumulative number of days the animals had been photoidentified previously) was only slightly less effective (ΔAIC = 1.4) (Table 2). In contrast, *api* (whether any animals had been photoidentified previously) performed considerably worse as a predictor of the proportion of poor photographs compared with *ppi* or *cpi* (ΔAIC = 44; Table 2).

The modelled proportion of poor photographs ($Q=1−3$) was $q_{13}=0.35$ (95%c.i. 0.30–0.41) for whales with no previous experience (*api* = 0) of the research vessel, falling to $q_{13}=0.30$ (95%c.i. 0.28–0.33) when at least one animal had experience (*api* = 1), $q_{13}=0.24$ (95%c.i. 0.21–0.27) when all animals had experience (*ppi* = 1), and $q_{13}=0.20$ (95%c.i. 0.17–0.24) when cumulative experience was at its highest (*cpi* = 84 animal days). These declines, which are statistically significant (at $P<0.05$; Table 2) are shown in Fig 3.

The declines in $q_{13}$ with all three independent variables are quite consistent within all primary study years (Fig 4; although data are unbalanced for 1999). Likewise, $q_{13}$ generally declined with exposure to the research vessel for individual social units that had been identified on at least five days within a year, with at least three members of the unit being identified on each day (Fig 5).

With median estimated range to the fluking whales as they were photographed ($r_{med}$) as a dependent variable, there was much less range in the AIC's for the different models than with $q_{13}$, indicating a less clear result (Table 2). The best fitting model included a decline in median range when any animals had previous experience of the research vessel, *api*. The decline, although statistically significant (at $P<0.05$), was not very steep, falling from 110m (95%c.i. 94-127m) when no animals had previous experience of the research vessel to 93m (95%c.i. 79-107m) when all had experience (Fig 6, Table 2).

When *api, ppi* and *cpi* were defined to just include previous identifications of each individual within the primary study year, excluding identifications in previous years, results were quite similar to the models when research-vessel exposure from previous years was included (S1 Supporting Information Table A1). While the *ppi* and *cpi* models fit better with the inclusion of experience from previous years, the model for *api* showed the reverse trend (ΔAIC = 3.48 (*ppi*), 2.22 (*cpi*), −11.26 (*api*); S1 Supporting Information Table A1). Thus, we do not seem to have sufficient data to decide whether experience of the research vessel had a consistent effect on whale behavior one or more years later.

## Discussion

The quadratic relationship between *sdd*, the number of days since the start of field season, and the efficiency of the photoidentification process could result from systematic changes in the photoidentification methods or crew abilities or whale behavior as field seasons progressed. Changes in whale behavior are more likely related to whale experience of the research vessel (covered by the dependent variables) than the length of the field season, and "new whales" were photographed at all stages of each field season. Thus, we suspect that the changes in phototidentification efficiency over the field season are more likely due to crew members learning in the early days, and becoming fatigued after several months.

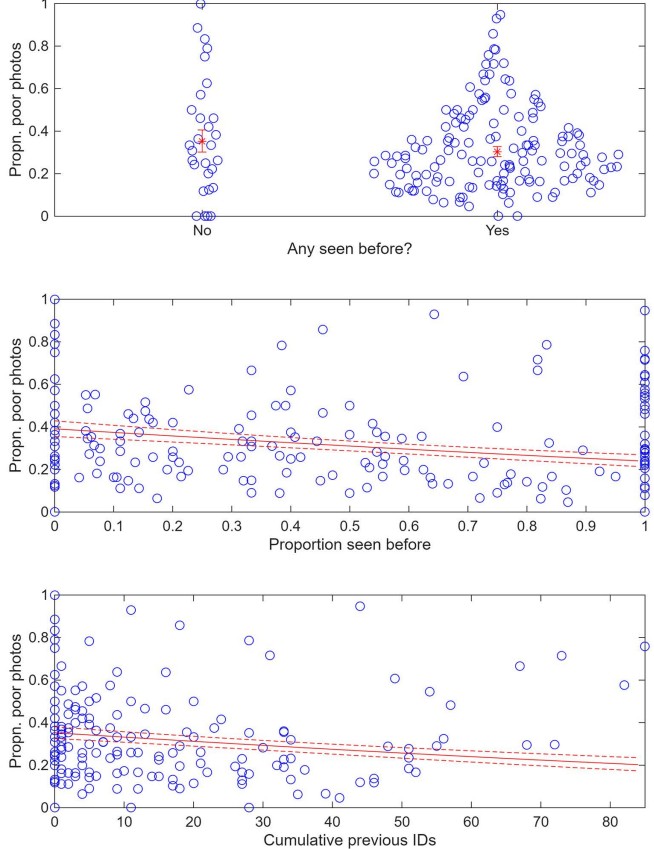

**Fig 3. Proportion of poor photographs ($q_{13}$) on each day (raw values) against the three independent variables (*api, ppi* and *cpi*).** Also shown are best fit estimates (plus 95% confidence intervals for the lines) from a generalized linear mixed effect model with binomial error.

The sperm whales off the Galápagos Islands became more tolerant of the photoidentification vessels with experience, allowing fewer poor photographs to be taken from slightly closer median ranges. This is a habituation-type response, "the relative persistent waning of a response" [4]. However, Bejder et al [6] noted that at least four different explanatory mechanisms could account for evidence of a habituation-type response (*i.e.,* reduced behavioral response over time): learning (individuals learn not to respond to the stimulus), displacement (the movement of less tolerant individuals away from the region of the stimulus), physiology (when repeated or prolonged exposure to that stimulus has caused physiological impairment, such as deafness), and ecological factors (such as effects of the disturbance on prey species). True habituation only occurs in the first of these scenarios. In our case, we studied the same individuals over time and used a research vessel with acoustic footprint far below those that cause physiological effects in cetaceans or effects on their prey [see 5]. Thus, we seem to have experienced true learned habituation by the followed sperm whales to an apparently benign research vessel.

Our results do not rule out the possibility that some individuals, or more probably social units, might not have been repeatedly followed due to their aversion to the research vessel. However, the research vessel typically detected (usually acoustically) and approached groups of sperm whales from ranges of several km, likely before the whales will have been aware of the vessel, and groups were followed fairly easily [18]. Thus, to minimize being tracked by the research vessel, social units would have to avoid the study area, which was large (tens of thousands of sq km) and had relatively little anthropogenic impact. Such displacement seems unlikely.

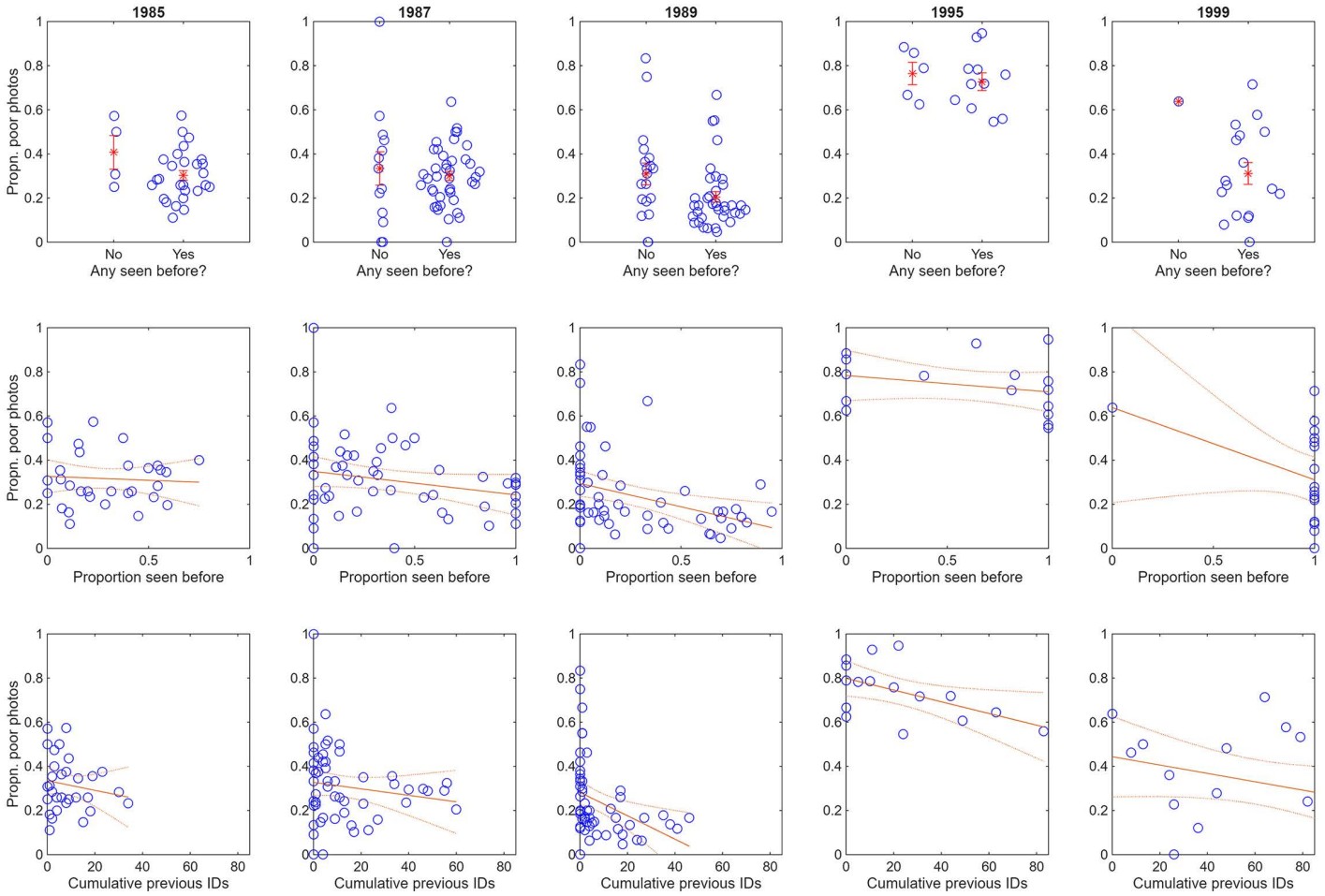

**Fig 4. Proportion of poor photographs ($q_{13}$) with the three independent variables (*api, ppi* and *cpi*) in each study year.** Also shown are best fit lines (plus 95% confidence intervals for the line) from generalized linear mixed effect models with binomial error. Data are quite few for 1995 and 1999, and poorly balanced for 1999.

In research on wild terrestrial animals, such as some primates, habituation can be very important [3]. Very much better behavioral information can be collected on animals that have been habituated to human presence. Thus, habituation may be a vital, and sometimes difficult to achieve, part of the research process [3]. There are parallels with the effects of our research process on sperm whales. Repeated experience of the research vessel made the sperm whales generally easier to photograph for the vital individually-identifying marks.

However, while the whales habituated to the presence of the research vessel, the research was not necessarily harm free. Tennessen et al. [24] found that vessel noise impairs foraging efficiency and success in killer whales (*Orcinus orca*). That seems less likely in our case because, the research vessel was low-powered, slow moving and the sperm whales typically feed several hundred metres beneath the surface and thus away from the sound source [25]. However, Gero and Whitehead [26] consider the possibility that the presence of a research vessel (one of those used in our habituation research) might have inhibited suckling by sperm whale calves. At a minimum, the whales seem to have decided the presence of the research vessel is less costly than modifying their behavior to avoid the research vessel (which was not the case for whaling [15]), but that does not mean there was no cost.

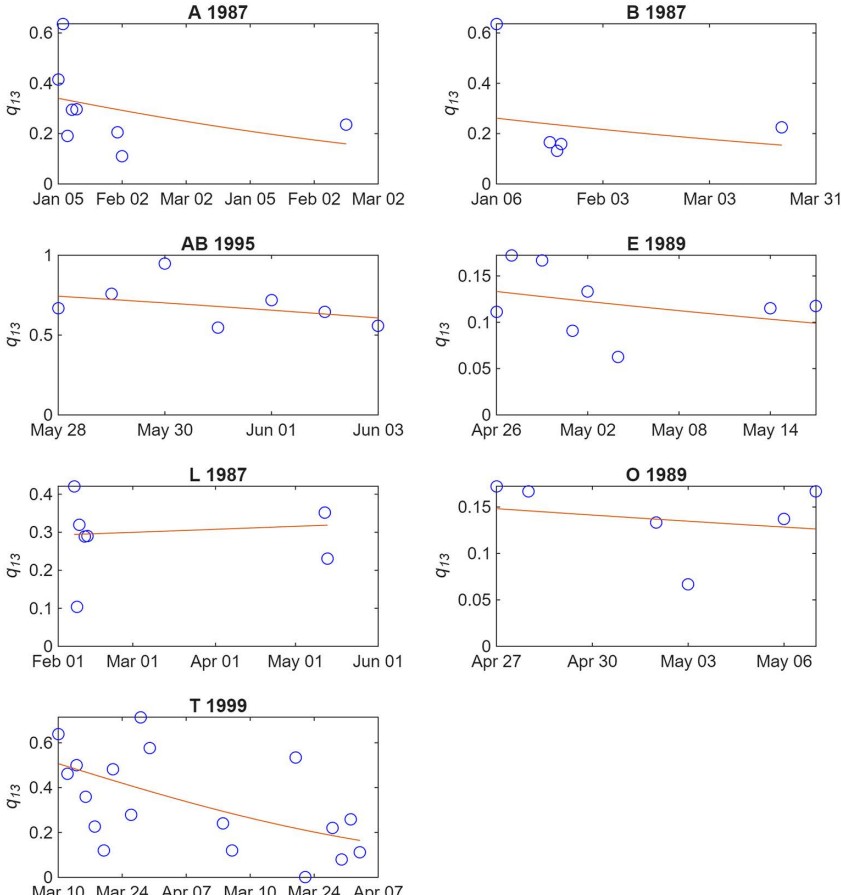

**Fig 5. Proportion of poor photographs ($q_{13}$) over time for days following social units that had been identified on at least 5 days within a year, with at least 3 members of the unit being identified on each day.** Also shown are best fit lines from generalized linear models with binomial error. Unit AB in 1995 is a merger of members of units A and B which were largely separate in 1987 [19].

Habituation to relatively harmless interactions with humans can result in reduced response to predators and less benign human intentions if individuals perceive these stimuli similarly [2]. For sperm whales, this may translate to an ameliorated response to other vessels which could present a risk of collision or entanglement with towed fishing gear. We suspect this is unlikely, as acoustic playback experiments show differential responses of sperm whales to predator and anthropogenic sounds [27]. However, considering the potential for habituation transfer in other contexts is critical to an accurate evaluation of the impacts of research on studied individuals.

Our data points (the proportion of poor photographs and median range to photographed animals on each day) have much variation so we cannot go too much past the primary result, that there is habituation to the presence of the research vessel. However, it does seem to be a gradual process—the more experience, the less aversion—rather than a brief experience by one or two animals reducing research-vessel aversion in the group as a whole (as evidenced by the general declines of the dependent variables with *ppi* and *cpi*, and the models including *ppi* or *cpi* fitting better than those with *api*). Also, we can say little as to whether the habituation has a social learning component, which was a factor in chimpanzee (*Pan troglodytes*) habituation in Samuni et al.'s [28] study, as well as in the development of the reaction of sperm whales to 19th Century whalers [15], or is purely the accumulated effect of individual learning among group members. Our

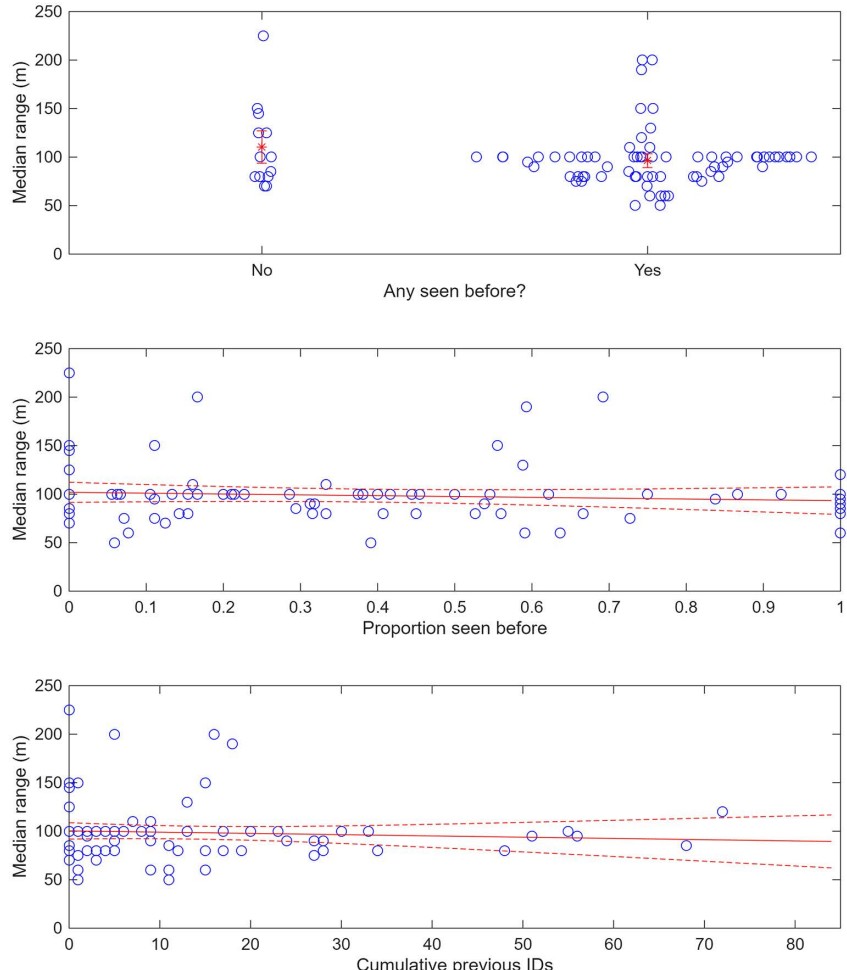

**Fig 6. Median range to photographed animals ($r_{med}$) on each day (raw values) against the three independent variables (*api, ppi* and *cpi*).** Also shown are best fit lines (plus 95% confidence intervals for the line) from generalized linear models.

results point towards experience of the research vessel in previous years having some bearing on habituation (as the best-fitting models of the proportion of poor photographs that included previous years' identifications had lower AIC than those that just used the focal year's identifications), but there is uncertainty (as the reverse was the case for *api*).

Social context (such as behavioral state, group size, presence of mature males or calves) could also affect measures of habituation. For example, the occurrence of evasive responses to human disturbances (drones) has been shown to increase with group size in belugas (*Delphinapterus leucas*) and guanacos (*Lama guanicoe*) [29,30].

If we extrapolate, the results indicate that photoidentification studies on sperm whales in new areas will be less effective than in places where the animals have experience of research vessels. We think it unlikely that the research is biased by habituation. Even though it was more difficult to obtain high quality photographs on days following naïve animals, we still individually identified many of the whales present. Furthermore, the habituation suggests that photoidentification research carried out in this manner is fairly benign, a reassuring conclusion for sperm whale scientists. However, this may depend on the research vessels behaving in the relatively restrained manner of those in our studies (including slow speeds, quiet engine, no biopsies, maintaining range to the animals). Field research on sperm

whales, and other cetaceans, is becoming more diverse, with field scientists using vessels that are often much faster than our auxiliary sailing vessel, often louder (for instance when using high-powered outboard motors), but sometimes quieter (electric motors [e.g., 31]). Techniques are changing, with photoidentifications and other data being obtained from drones and in other ways [e.g., 32]. Such field techniques will likely have quite different impacts on the animals than those assessed here.

Field scientists should examine whether animals do habituate to their persistent research techniques, enabling informed ethical reviews [see 1]. Although there have been studies of the short- and long-term impacts of specific instantaneous or nearly-instantaneous research techniques on cetaceans [e.g., 7–9], the effects of repeated exposures have received almost no consideration. For ethical field research, the benefits of the research to the animals or their habitat should outweigh adverse impacts of the research process itself.

This study used old data, and poor photographs, that, depending on the circumstances or study species, can indicate evasiveness, or comfort with the research process (e.g., "relaxed" dolphins that do not show their fins). Old data are often ignored, and poor photographs discarded, but in this study they proved invaluable. They can be useful data sources for behavioral research and other kinds of study (such as changes in epidermal markings over time).

## Supporting information

**S1 Dataset. Data for analysis in Excel file. Sheets: All previous years; Just year in question; Data for social units; Explanation.**
(PDF)

**S1 Supporting Information. Data description; Additional dependent variables.** Fig A1 Dependent variables plotted against days since the start of that year's field study; Fig A2 Proportion of very poor photographs on each day against the three independent variables; Fig A3 Mean range to photographed animals on each day against the three independent variables; Table A1 With the proportion of poor photographs as the dependent variable, compares the fit of models to independent and control variables when the independent variables were calculated using prior identifications from just the current year or current and previous years.
(PDF)

## Acknowledgments

We thank the crew who collected the data at sea, as well as Tom Arnbom, Susan Waters, Susan Dufault, Jenny Christal and Luke Rendell who analyzed the photoidentifications. We are particularly grateful to the late Godfrey Merlen for his ongoing photoidentification work off the Galápagos Islands, as well as logistical and practical assistance to our field studies. Field work was carried out under the auspices of the Charles Darwin Research Station, with the permission of the Galápagos National Parks Service, the Instituto Nacional de Pesca, and the Armada of Ecuador. We thank Ted Cheeseman and an anonymous reviewer for constructive comments on the manuscript.

## Author contributions

**Conceptualization:** Hal Whitehead.

**Data curation:** Ana Eguiguren.

**Formal analysis:** Hal Whitehead.

**Funding acquisition:** Hal Whitehead.

**Methodology:** Hal Whitehead, Christine M. K. Clarke.

**Resources:** Ana Eguiguren.

**Writing – original draft:** Hal Whitehead.

**Writing – review & editing:** Hal Whitehead, Christine M. K. Clarke, Ana Eguiguren.

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
