## [Decision Letter · Decision Letter 0]

25 Feb 2026

PONE-D-25-53637Sperm whales habituate to research vessels engaged in photoidentificationPLOS One

Dear Dr. Whitehead,

Thank you for submitting your manuscript to PLOS ONE. After careful consideration, we feel that it has merit but does not fully meet PLOS ONE’s publication criteria as it currently stands. Therefore, we invite you to submit a revised version of the manuscript that addresses the points raised during the review process. Apologies for the delay in providing you with a decision on this manuscript. A large number of invited reviewers declined, but I was finally able to secure two suitable reviewers, who have both provided constructive and positive feedback on the manuscript.

We look forward to receiving your revised manuscript.

Kind regards,

Joel Harrison Gayford

Academic Editor

PLOS One

Journal Requirements:

“Natural Sciences and Engineering Research Council of Canada,

Whale and Dolphin Conservation Society,

National Geographic Society,

International Whaling Commission,

Cetacean Society International,

Green Island Foundation.”

3. We note that Figures 1 and 2 in your submission contain copyrighted images. All PLOS content is published under the Creative Commons Attribution License (CC BY 4.0), which means that the manuscript, images, and Supporting Information files will be freely available online, and any third party is permitted to access, download, copy, distribute, and use these materials in any way, even commercially, with proper attribution. For more information, see our copyright guidelines: http://journals.plos.org/plosone/s/licenses-and-copyright.

(1)       You may seek permission from the original copyright holder of Figures 1 and 2 to publish the content specifically under the CC BY 4.0 license.

5. We note that there is identifying data in the Supporting Information file Habituaton_data. Due to the inclusion of these potentially identifying data, we have removed this file from your file inventory. Prior to sharing human research participant data, authors should consult with an ethics committee to ensure data are shared in accordance with participant consent and all applicable local laws.

-Location data

Please remove or anonymize all personal information Habituaton_data, ensure that the data shared are in accordance with participant consent, and re-upload a fully anonymized data set. Please note that spreadsheet columns with personal information must be removed and not hidden as all hidden columns will appear in the published file.

Reviewers' comments:

Reviewer's Responses to Questions

**Comments to the Author**

1. Is the manuscript technically sound, and do the data support the conclusions?

Reviewer #1: Yes

Reviewer #2: Yes

2. Has the statistical analysis been performed appropriately and rigorously? 

Reviewer #1: Yes

Reviewer #2: Yes

3. Have the authors made all data underlying the findings in their manuscript fully available?

Reviewer #1: Yes

Reviewer #2: Yes

4. Is the manuscript presented in an intelligible fashion and written in standard English?

Reviewer #1: Yes

Reviewer #2: Yes

5. Review Comments to the Author

Reviewer #1: I have carefully read the manuscript titled “Sperm whales habituate to research vessels engaged in photoidentification”. The authors present a resourceful and innovative analysis of historical data to address the ethical and practical impacts of photoidentification on sperm whales, a topic that has to date has not received attention despite the technique's ubiquity. The authors’ approach is particularly clever. Rather than discarding low-quality photographs, which is common in this type of capture-recapture studies, they repurpose these "failures" as a proxy for behavioral avoidance. By quantifying the proportion of poor-quality images and correlating them with range estimates, the study creates a metric for ‘evasiveness’ without requiring separate, dedicated behavioral sampling tracks. This retrospective use of archival data sets a valuable precedent for other long-term monitoring projects to assess their own observer effects without the need for new field interventions. I recommend a minor revision, with a few suggestions related to data representation below.

The methodological rigor is generally strong, employing Generalized Linear Mixed Models (GLMMs) that appropriately control for Year as a random variable and ‘days since start of season’ (sdd) to account for systematic changes in crew ability or whale behavior. The distinction made between ‘proportion of previously identified animals’ (ppi) and ‘cumulative experience’ (cpi) allows for a nuanced understanding of how exposure drives tolerance, with the results suggesting a linear decrease in avoidance behavior as familiarity increases. However, the manuscript could be improved by offering a deeper interpretation of the control variable sdd, which showed a U-shaped quadratic effect where poor photographs decreased for the first 80 days but rose toward the end of the season. While the authors statistically control for this, the Discussion lacks a biological or operational hypothesis for this late-season reversal—potentially addressing whether this indicates observer fatigue or a shift from habituation to sensitization after prolonged exposure.

Regarding the interpretation of results, the authors successfully argue for true habituation rather than displacement or physiological impairment, noting that the acoustic footprint of their sailing vessels was likely too low to cause physical damage. However, the discussion regarding ‘habituation transfer’ to other vessel types warrants expansion. Given that the data set concludes in 1999 and relies on auxiliary sailing vessels, the authors should discuss how these findings translate into modern research environments involving high-speed motorized vessels or drones, which may elicit different behavioral responses. Explicitly contrasting the ‘benign’ nature of their specific vessel protocols (slow approach, no biopsy) against more invasive modern techniques would strengthen the external validity of their conclusion that photo-ID is largely harmless.

Overall, this is a scientifically sound contribution that validates the ethics of long-term field studies, confirming that sperm whales likely learn to tolerate researchers rather than suffering physiological impairment. The study transforms a potential methodological weakness (poor photography) into a useful analytical tool, demonstrating that "failed" data points can contain vital information about animal welfare.

Specific suggestions for figures and data representation:

Figure 1 and Figure 2

Figure 1 (the vessel following a whale) and Figure 2 (examples of Q-values) would be more effective if combined into a single ‘Methods Plate’. Currently, Figure 2 displays photographic quality ranging from Q=2 to Q=5. However, the study’s primary dependent variable is q13 , the proportion of ‘poor’ photographs defined as Q = 1–3. Figure 2 should include an example of a Q=1 (‘extremely poor’) photograph to aid the reader in visually understanding what constitutes an ‘evasive’ or ‘failed’ data point. Furthermore, the caption for the vessel image (Figure 1) should explicitly state the distance and orientation shown (e.g., 50m behind, matching speed) to visually confirm the ‘benign’ protocol described in the text.

Figures 3 and 6

For Figures 3 and 6, which display the effects of experience (ppi, api, cpi) on photo quality (q13) and range (rmed), the presentation of the variable needs adjustment. The text defines as a binary measure (whether any animals had been identified previously: yes/no). Plotting a continuous regression line with confidence intervals for a binary variable can be visually misleading. It would be more rigorously presented as a box-and-whisker plot or a violin plot comparing the two states (Experience vs. No Experience). Additionally, for the continuous variables (ppi and cpi), the authors should clarify in the captions whether the data points plotted are raw values or partial residuals controlled for the ‘Year’ and ‘day of season’ (sdd) variables, as the text notes that has a non-linear U-shaped effect that was statistically controlled for.

Figure 4

The figure caption should acknowledge that the wide confidence intervals in 1995 and 1999 indicate low predictive power due to small sample sizes.

The regression line for the 1995 and 1999 panels are visually weak due to massive confidence intervals. The text claims results are ‘quite consistent within all primary study years’, but Figure 4 visually contradicts this for 1995/1999 by showing lack of precision. The authors should soften the claim of consistency or explicitly note the increased uncertainty shown in these specific panels.

Supplementary Figure S1

The text describes a "U-shaped quadratic effect" where photo quality initially improves but then deteriorates after 80 days into a season. Given that this finding complicates the ‘habituation’ narrative (potentially suggesting observer fatigue or late-season sensitization), the authors should consider moving Figure S1 from the Supplementary Materials into the main manuscript. Visualizing this reversal is important for the transparency of the study, as it justifies the inclusion of the term in the final statistical models.

Reviewer #2: This manuscript uses old (pre-digital camera) photo-ID based data to examine whether sperm whales (Physeter macrocephalus) habituate to research vessels conducting surveys off the Galápagos Islands. Using data collected across five primary study years (1985–1999), the authors show that whales previously exposed to the research vessel yielded higher-quality photographs taken at shorter median ranges — interpreted as evidence of habituation.

By my judgement the manuscript is publication-ready or nearly so, the exceptions being minor, discussion points that I believe will add value for readers. None of these are critical:

1 - Lines 199-208, first paragraph of Discussion: of explanatory mechanisms for habituation-type response, the authors are quick to dismiss mechanisms other than a learning-based habituation response on grounds that the individuals in the study have been re-sighted over many years, and a quiet boat. This feels legit to a reader familiar with Whitehead et al’s long history of work with this population but I feel deserves stronger support, ie how do we known that resight was not biased to individuals amenable to habituation?

2 - Understood the data have a great deal of variation limiting power to interpret multi-year habituation but is there an opportunity for discussion of inter- vs intra-year habituation? Is there any indication of de-habituation between seasons? Can anything be learned about the habituation process, ie individuals become more tolerant of one well behaved vessel during each season but are behaviors changing over lifetimes of these animals? I’d think Gero’s work in Dominica has seen whether or not this is so, though am not aware of if there’s any publications on the topic.

3 - Could the research be biased by habituation? Better quality photos of (presumably) more relaxed, less behavior state-shifting family groups is likely a positive but are there any blindspots that we are missing?

Very minor edits:

Line 126-135 - seems logical to reorder to api -> ppi -> cpi to scalę from simple to complex measure of familiarity (here and in subsequent mentions of the three metrics)

Line 235 - add comma: the more experience, the less aversion

6. PLOS authors have the option to publish the peer review history of their article (what does this mean?). If published, this will include your full peer review and any attached files.

Reviewer #1: No

Reviewer #2: **Yes:** Dr. Ted Cheeseman

---

## [Author Response · Author response to Decision Letter 1]

12 Mar 2026

Uploaded "Response to Reviewers" file

---

## [Decision Letter · Decision Letter 1]

6 Apr 2026

PONE-D-25-53637R1Sperm whales habituate to research vessels engaged in photoidentificationPLOS One

Dear Dr. Whitehead,

Thank you for submitting your manuscript to PLOS ONE. After careful consideration, we feel that it has merit but does not fully meet PLOS ONE’s publication criteria as it currently stands. Therefore, we invite you to submit a revised version of the manuscript that addresses the points raised during the review process.

We look forward to receiving your revised manuscript.

Kind regards,

Joel Harrison Gayford

Academic Editor

PLOS One

Journal Requirements:

Reviewers' comments:

Reviewer's Responses to Questions

**Comments to the Author**

1. If the authors have adequately addressed your comments raised in a previous round of review and you feel that this manuscript is now acceptable for publication, you may indicate that here to bypass the “Comments to the Author” section, enter your conflict of interest statement in the “Confidential to Editor” section, and submit your "Accept" recommendation.

Reviewer #1: All comments have been addressed

Reviewer #2: All comments have been addressed

2. Is the manuscript technically sound, and do the data support the conclusions?

Reviewer #1: Yes

Reviewer #2: Yes

3. Has the statistical analysis been performed appropriately and rigorously? 

Reviewer #1: Yes

Reviewer #2: Yes

4. Have the authors made all data underlying the findings in their manuscript fully available?

Reviewer #1: Yes

Reviewer #2: Yes

5. Is the manuscript presented in an intelligible fashion and written in standard English?

Reviewer #1: Yes

Reviewer #2: Yes

6. Review Comments to the Author

Reviewer #1: Good work addressing the reviewer comments.The manuscript at its current version is improved and should be a strong contribution to the literature.

Reviewer #2: satisfied with revisions and explanations. I do not have any further comments at this time. Have a nice day!

7. PLOS authors have the option to publish the peer review history of their article (what does this mean?). If published, this will include your full peer review and any attached files.

Reviewer #1: No

Reviewer #2: **Yes:** Ted Cheeseman

---

## [Author Response · Author response to Decision Letter 2]

18 Apr 2026

In this iteration we received no new feedback and that all previous editorial concerns have been addressed. See email exchange with Kit Stokes, Associate Peer Review Operations Specialist.

---

## [Editor Report · Decision Letter 2]

21 Apr 2026

Sperm whales habituate to research vessels engaged in photoidentification

PONE-D-25-53637R2

Dear Dr. Whitehead,

We’re pleased to inform you that your manuscript has been judged scientifically suitable for publication and will be formally accepted for publication once it meets all outstanding technical requirements.

Kind regards,

Joel Harrison Gayford

Academic Editor

PLOS One
---

## [Editor Report · Acceptance letter]

PONE-D-25-53637R2

PLOS One

Dear Dr. Whitehead,

I'm pleased to inform you that your manuscript has been deemed suitable for publication in PLOS One. Congratulations! Your manuscript is now being handed over to our production team.

Kind regards,

on behalf of

Mr. Joel Harrison Gayford

Academic Editor

PLOS One